# Reactive oxygen species regulate activity-dependent neuronal plasticity in *Drosophila*

**Matthew CW Oswald[1]\*, Paul S Brooks[1], Maarten F Zwart[2], Amrita Mukherjee[1], Ryan JH West[3,4], Carlo NG Giachello[3], Khomgrit Morarach[1], Richard A Baines[3], Sean T Sweeney[4†]\*, Matthias Landgraf[1†]\***

[1]Department of Zoology, University of Cambridge, Cambridge, United Kingdom; [2]HHMI Janelia Research Campus, Ashburn, United States; [3]Faculty of Biology, Medicine and Health, University of Manchester, Manchester, United Kingdom; [4]Department of Biology, University of York, York, United Kingdom

**Abstract** Reactive oxygen species (ROS) have been extensively studied as damaging agents associated with ageing and neurodegenerative conditions. Their role in the nervous system under non-pathological conditions has remained poorly understood. Working with the *Drosophila* larval locomotor network, we show that in neurons ROS act as obligate signals required for neuronal activity-dependent structural plasticity, of both pre- and postsynaptic terminals. ROS signaling is also necessary for maintaining evoked synaptic transmission at the neuromuscular junction, and for activity-regulated homeostatic adjustment of motor network output, as measured by larval crawling behavior. We identified the highly conserved Parkinson's disease-linked protein DJ-1β as a redox sensor in neurons where it regulates structural plasticity, in part via modulation of the PTEN-PI3Kinase pathway. This study provides a new conceptual framework of neuronal ROS as second messengers required for neuronal plasticity and for network tuning, whose dysregulation in the ageing brain and under neurodegenerative conditions may contribute to synaptic dysfunction.
DOI: https://doi.org/10.7554/eLife.39393.001

**\*For correspondence:**
mo364@cam.ac.uk (MCWO);
sean.sweeney@york.ac.uk (STS);
ml10006@cam.ac.uk (ML)

[†]These authors contributed equally to this work

**Competing interests:** The authors declare that no competing interests exist.

## Introduction

Levels of reactive oxygen species (ROS) in the brain increase with ageing and high levels of ROS are a hallmark of neurodegeneration, including Alzheimer's and Parkinson's disease (*Höhn and Grune, 2013*; *Martins et al., 1986*; *Spina and Cohen, 1989*) for review see (*Milton and Sweeney, 2012*). Mitochondria are a significant source of ROS, which form as obligate byproducts of respiratory ATP synthesis by 'leakage' of the electron transport chain, thus leading to the generation of superoxide anions ($O_2^-$) and hydrogen peroxide ($H_2O_2$) (*Halliwell, 1992*). Implicit in their name, ROS are highly reactive, containing one or more unpaired electrons, with the potential to modify and damage by oxidation proteins, lipids and DNA (*Gladyshev, 2014*; *Harman, 1956*; *Stuart et al., 2014*). Importantly, ROS have also been recognized as signaling molecules in metabolic pathways (*Liemburg-Apers et al., 2015*) and controlling the activity of transcription factors such as AP-1 and Nrf2 (*Jindra et al., 2004*; *Soriano et al., 2009*). Moreover, several kinase signaling pathways are enhanced by ROS, either by oxidation of kinase interacting modulators, such as thioredoxin or glutathione-S-transferases (*Adler et al., 1999*; *Saitoh et al., 1998*), or through inhibition of counteracting phosphatases, for example PTEN, by oxidation of the active site cysteine residue (*Finkel, 2011*; *Stuart et al., 2014*; *Tonks, 2005*).

We previously showed in a model for lysosomal storage diseases that ROS can regulate neuro-muscular junction (NMJ) structure (*Milton et al., 2011*). NMDA receptor stimulation can lead to ROS

generation (*Bindokas et al., 1996*; *Brennan et al., 2009*; *Dugan et al., 1995*), and in hippocampal and spinal cord slices ROS have been shown sufficient and necessary for inducing 'Hebbian' forms of plasticity (LTP) (*Kamsler and Segal, 2003a*; *Kamsler and Segal, 2003b*; *Klann, 1998*; *Knapp and Klann, 2002*; *Lee et al., 2010*). Conversely, disturbing the ROS balance by over-expression of the scavenger superoxide dismutase caused defects in hippocampal LTP and learning paradigms in mice (*Gahtan et al., 1998*; *Levin et al., 1998*; *Thiels et al., 2000*). Recent studies have linked increased ROS levels with neurodevelopmental conditions such as schizophrenia, bipolar and autism spectrum disorders (*Do et al., 2015*; *Steullet et al., 2017*).

Here, we set out to investigate potential roles for ROS in the nervous system under non-pathological conditions, which are much less well understood. The brain is arguably the most energy demanding organ and mitochondrial oxidative phosphorylation is a major source of ROS (*Attwell and Laughlin, 2001*; *Hallermann et al., 2012*; *Zhu et al., 2012*). We therefore asked whether neurons might utilize mitochondrial metabolic ROS as feedback signals to mediate activity-regulated changes. As an experimental model we used the motor system of the fruitfly larva, *Drosophila melanogaster*, which allows access to uniquely identifiable motoneurons in the ventral nerve cord and their specific body wall target muscles (*Kohsaka et al., 2012*). We established an experimental paradigm for studying activity-regulated structural adjustments across an identified motoneuron, quantifying changes at both pre- and postsynaptic terminals. We show that thermogenetic neuronal over-activation leads to the generation of ROS at presynaptic terminals, and that ROS signaling is necessary and sufficient for the activity-regulated structural adjustments. As a cellular ROS sensor we identified the conserved redox sensitive protein DJ-1β, a homologue of vertebrate DJ-1 (PARK7) (*Meulener et al., 2005*), and the phosphatase and tensin homolog (PTEN) and PI3kinase as downstream effectors of activity-ROS-mediated structural plasticity. We find that ROS signaling is also required for maintaining constancy of evoked transmission at the neuromuscular junction (NMJ) with a separate ROS pathway regulating the amplitude of spontaneous vesicle release events. Behaviourally, ROS signaling is required for the motor network to adjust homeostatically to return to a set crawling speed following prolonged overactivation.

In summary, this study establishes a new framework for studying ROS in the nervous system: as obligatory regulators that inform neurons about their activation status, and as obligatory mediators of activity-induced plasticity, both structural and physiological.

## Results

### Structural plasticity of synaptic terminals is regulated by neuronal activity

Our aim was to explore roles for activity-regulated ROS signaling in the nervous system under non-pathological conditions. Working with the *Drosophila* larval neuromuscular system allowed us to target manipulations to identified nerve cells in vivo that manifest structural and functional plasticity (*Frank et al., 2013*; *Tripodi et al., 2008*; *Wolfram and Baines, 2013*; *Zwart et al., 2013*). We focused on two well characterized motoneurons, 'aCC' and 'RP2', which jointly innervate the dorsal acute muscle 1 (DA1) (*Figure 1A*) (*Baines et al., 1999*; *Baines et al., 2001*; *Bate, 1993*; *Choi et al., 2004*; *Hoang and Chiba, 2001*; *Landgraf et al., 2003*; *Sink and Whitington, 1991*). First, we characterized activity-regulated morphological changes at the presynaptic neuromuscular junction (NMJ) and, in the central nervous system (CNS), the branched postsynaptic dendritic arbors that receive input from premotor interneurons (*Baines et al., 1999*; *Schneider-Mizell et al., 2016*; *Zwart et al., 2013*). A simple method for increasing activity in the larval locomotor network is to increase ambient temperature from the default standard of 25°C to 29°C or 32°C. *Sigrist et al. (2003)* and *Zhong and Wu (2004)* previously demonstrated this to trigger increased locomotor activity and to result in increased varicosity (bouton) number at presynaptic NMJs (*Sigrist et al., 2003*; *Zhong and Wu, 2004*). We were able to reproduce and extend these findings: rearing larvae at higher ambient temperatures results in NMJs having more boutons than controls by the third instar wandering stage, 100 hr after larval hatching (ALH) (grey data in *Figure 1B–C*, compare 25°C with 29°C and 32°C conditions). To complement these systemic manipulations and to exclude potential non-specific effects we turned to a cell-specific activation paradigm of selectively overactivating the aCC and RP2 motoneurons via targeted mis-expression of the warmth-gated cation channel dTrpA1. Expression of

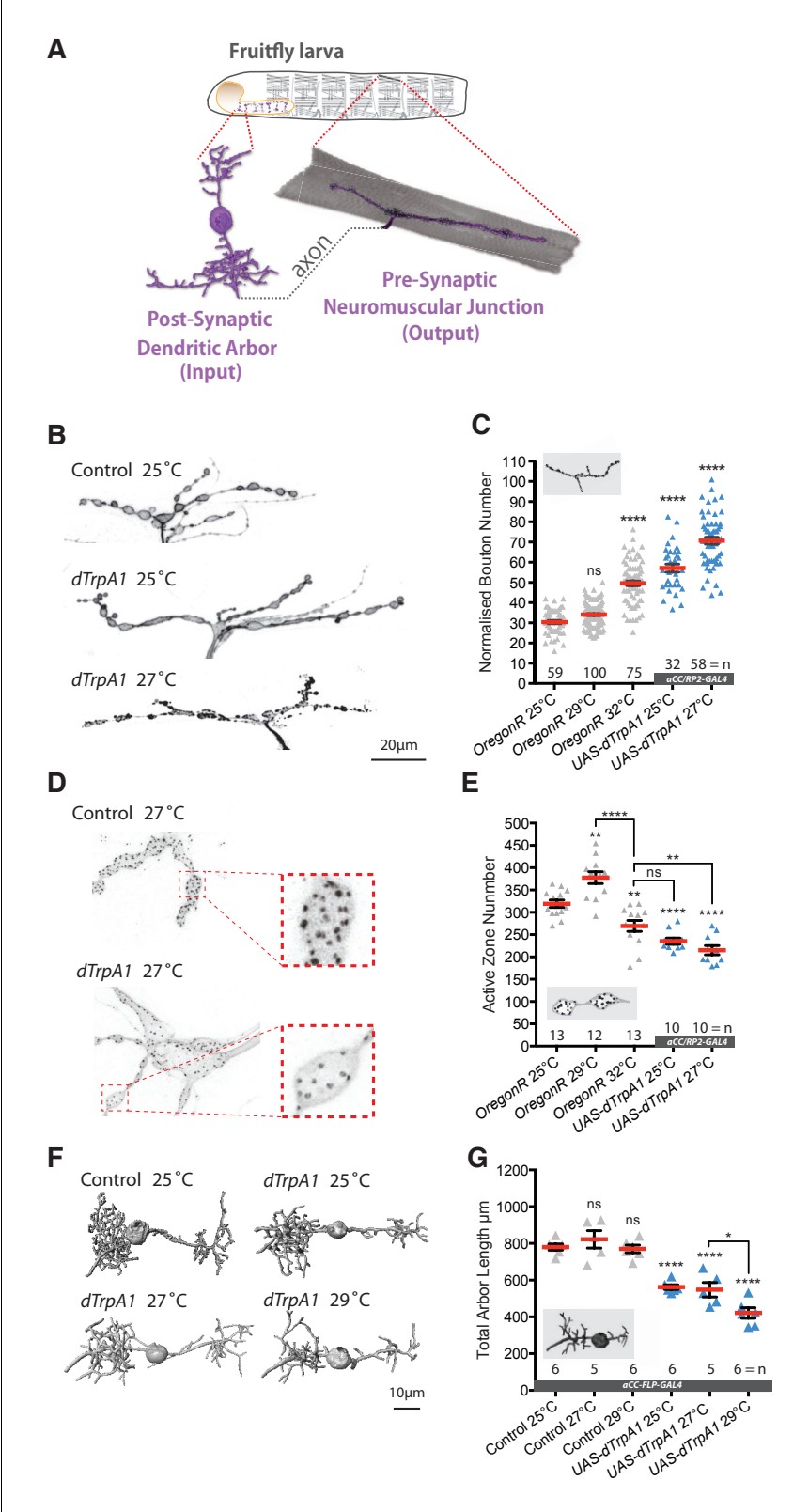

**Figure 1.** Adaptive structural synaptic plasticity at motoneuron input and output terminals in response to increased neuronal activity. (**A**) Graphical illustration of a stereotypical larval motoneuron (MN) (adapted from *Kohsaka et al., 2012*). Pre-motor interneurons make synaptic connections with the MN dendritic arbor (input) in the larval ventral nerve cord (equivalent of mammalian spinal cord). The MN extends an axonal projection into the periphery where it connects with a target muscle via an NMJ, characterized by varicose swellings (boutons) each containing multiple individual

*Figure 1 continued on next page*

*Figure 1 continued*

neurotransmitter release sites (active zones). (**B and C**) Representative images of muscle DA1 [muscle one according to (***Crossley, 1978***)] NMJs from third instar larvae (100 hr ALH). Dot-plot quantification shows NMJ bouton number increases in response to systemic and cell-specific activity increases. (**D and E**) Active zone number increases following low-level overactivation (29°C), but progressively reduces upon stronger overactivation. (**F and G**) Digital reconstructions and dot plots show that overactivation leads to reduced total dendritic arbor length of aCC motoneurons (24 hr ALH). '*aCC/RP2-GAL4*' expresses GAL4 in all, '*aCC-FLP-GAL4*' in single aCC and RP2 motoneurons (see Methods for details); 'Control' in (**B–G**) is heterozygous *aCC/RP2-GAL4* or *aCC-FLP-GAL4*, achieved by crossing the respective GAL4 line to Oregon-R wild type. Mean ± SEM, ANOVA, ns = not significant, *p<0.05, **p<0.01, ***p<0.001, ****p<0.0001, n = replicate number. Comparisons with control are directly above data points.

DOI: https://doi.org/10.7554/eLife.39393.002

The following source data is available for figure 1:

**Source data 1.** Source data for *Figure 1*.

DOI: https://doi.org/10.7554/eLife.39393.003

dTrpA1 in neurons is a well established method for temperature controlled neuronal overactivation (***Hamada et al., 2008***; ***Owald et al., 2015***). For larval motoneurons in particular, Pulver and colleagues demonstrated that dTrpA1 is activated above 24°C, at 25°C leading to action potential firing frequencies of 9–12 Hz (moderate activation) and 22–30 Hz at 27°C (stronger activation) (***Pulver et al., 2009***). These activation levels are within the physiological range of larval motoneurons, thought to operate at approximately 42 Hz during muscle contraction cycles (***Chouhan et al., 2010***). Similar to the systemic manipulations, we found that cell-specific thermogenetic dTrpA1 activation of single motoneurons also led to titratable increases in bouton number at presynaptic NMJs (blue data in *Figure 1B–C*). Note that these cell-specific dTrpA1-mediated activity manipulations were carried out at 25°C and 27°C, sufficient to activate dTrpA1 expressing neurons, but otherwise not causing significant changes in NMJ morphology in non-expressing motoneurons (***Tsai et al., 2012***).

Next, we quantified synapse number at the presynaptic NMJ on muscle DA1, measured by active zones (visualized with the nc82 antibody against the active zone protein Bruchpilot (***Wagh et al., 2006***)). This describes a more complex relationship. As previously published, a moderate increase in activity, for example by rearing larvae at 29°C, causes both more boutons and also more active zones to be formed, potentiating transmission at the NMJ (***Sigrist et al., 2003***). In contrast, further increases in network activity, as effected by rearing larvae at 32°C or by cell-specific dTrpA1-mediated motoneuron activation at increasing temperatures led to progressive active zone reductions (*Figure 1D,E*).

We then looked at activity-regulated structural changes of the postsynaptic dendritic arbor of the aCC motoneuron, which is known to be plastic during embryonic and larval stages (***Hartwig et al., 2008***; ***Tripodi et al., 2008***). To this end, we targeted GAL4 and dTrpA1 expression to individual aCC motoneurons (***Ou et al., 2008***). Morphometric analysis revealed that the size of the aCC postsynaptic dendritic arbor decreased with rising levels of temperature-gated dTrpA1 activity (*Figure 1F,G*). We and others previously showed that dendritic length of these neurons correlates with input synapse number and synaptic drive (***Schneider-Mizell et al., 2016***; ***Zwart et al., 2013***).

In summary, we find that the synaptic terminals of larval motoneurons undergo titratable structural changes in response to neuronal overactivation. Postsynaptic dendritic arbor size, and by inference synapse number and synaptic drive (***Zwart et al., 2013***), negatively correlate with activation levels. At the presynaptic NMJ, synapse number also correlates negatively with activation level - bar a narrow low level activity window that can lead to potentiation (***Ataman et al., 2008***; ***Piccioli and Littleton, 2014***; ***Sigrist et al., 2003***). Boutons provide an additional anatomical readout for NMJ plasticity, increasing in number with levels of activity. However, no functional significance of bouton number and size has been documented and changes in these bouton parameters are not predictive of changes in synaptic transmission (***Campbell and Ganetzky, 2012***).

## Neuronal overactivation leads to ROS generation in presynaptic terminals

Next, we asked if in vivo overactivation of individual motoneurons is associated with increased ROS levels, as reported for hippocampal neurons in culture (***Hongpaisan et al., 2004***). To this end, we co-expressed in aCC and RP2 motoneurons the mitochondrion-targeted ratiometric ROS reporter

*UAS-mito-roGFP2-Orp1* (*Gutscher et al., 2009*) along with *UAS-dTrpA1*. We focused on mitochondria at the muscle DA1 NMJs of wandering third instar larvae (100 hr ALH). Quantification shows a clear trend of increasing temperature and dTrpA1-mediated activation resulting in progressively greater mean oxidation levels of this ROS sensor in mitochondria at the NMJ (*Figure 2A*). These data show that in vivo dTrpA1-mediated overactivation of *Drosophila* larval motoneurons leads to increased mitchdondrial ROS at presynaptic NMJs.

## Activity generated ROS regulate structural plasticity at synaptic terminals

We then tested whether or not ROS are required for activity-dependent structural synaptic terminal plasticity. To this end we increased neuronal activity in the aCC and RP2 motoneurons via dTrpA1 expression, rearing larvae at 25°C, the lower threshold of dTrpA1 activation that leads to activity-dependent changes of synaptic terminals. At the same time we additionally over-expressed in these neurons ROS scavenging enzymes: Superoxide Dismutase 2 (SOD2), which catalyses $O_2^-$ to $H_2O_2$ reduction; or Catalase, which converts $H_2O_2$ into $H_2O$ and $O_2$. Over-expression of either ROS scavenger enzyme alone showed no effect on NMJ bouton number (*Figure 2B*). For Catalase over-expression we further quantified active zone number at the NMJ and dendritic arbor size and there too found this indistinguishable from controls (*Figures 2C* and *3B*). In contrast, co-expression of *UAS-Catalase* with *UAS-dTrpA1* significantly counteracted changes in bouton and active zone number otherwise caused by dTrpA1-mediated neuronal overactivation (25°C) (*Figures 2B* and *3B*). Similarly, at the motoneuron input terminals in the CNS, Catalase co-expression rescued

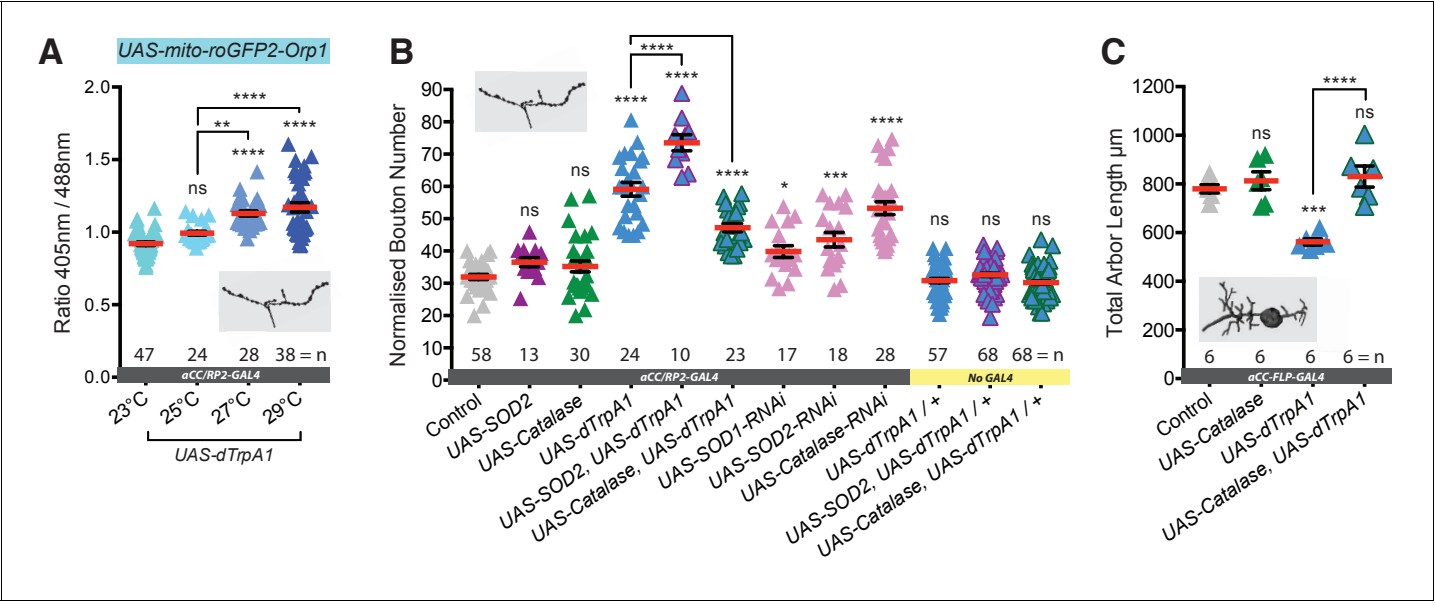

**Figure 2.** Neuronal activation leads to generation of synaptic ROS that regulate structural plasticity at input and output terminals. (**A**) Elevated cell-specific activity increases mitochondrial ROS production at NMJs. Dot plots of mitochondrion-targeted ratiometric $H_2O_2$ sensor (*UAS-mito-roGFP2-Orp1* [*Gutscher et al., 2009*]) in wandering third instar larval NMJs (100 hr ALH) at 23°C (control, dTrpA1 inactive), 25°C (moderate), 27°C (strong) and 29°C (very strong) dTrpA1-mediated overactivation. (**B**) Bouton number at the NMJ is increased by *UAS-dTrpA1*-mediated overactivation. This is exacerbated by co-expression of *UAS-SOD2* (converts $O_2^-$ to $H_2O_2$) and rescued by expression of the $H_2O_2$ scavenger *UAS-Catalase*. Cell-specific ROS elevation by scavenger knockdown is sufficient to induce NMJ elaboration. *aCC/RP2-GAL4*, 'Control' is *aCC/RP2-GAL4* alone. (**C**) Total dendritic arbor length is reduced by single cell overactivation, but rescued by co-expression of the $H_2O_2$ scavenger *UAS-Catalase* (aCC motoneurons, 24 hr ALH). *aCC-FLP-GAL4*, 'Control' is *aCC-FLP-GAL4* alone, heterozygous, achieved by crossing the GAL4 driver to Oregon-R wild type flies. Larvae were reared at 25°C unless indicated otherwise. Mean ± SEM, ANOVA, ns = not significant, *$p<0.05$, **$p<0.01$, ***$p<0.001$, ****$p<0.0001$, n = replicate number..
DOI: https://doi.org/10.7554/eLife.39393.004

The following source data is available for figure 2:

**Source data 1.** Source data for *Figure 2*.
DOI: https://doi.org/10.7554/eLife.39393.005

dendritic arbor size (*Figure 2C*). Conversely, SOD2 co-expression enhanced the dTrpA1-mediated increase of bouton number at the NMJ, presumably by potentiating conversion of activity-generated increase of $O_2^-$ into $H_2O_2$ (*Figure 2B*). We next asked if increasing neuronal ROS was sufficient to invoke structural plasticity in the absence of dTrpA1 activity manipulations. Indeed, cell-specific RNAi knock down of any one of three endogenous ROS scavengers, SOD1, SOD2 or Catalase, led to NMJs with increased bouton number, in agreement with a prior study by *Milton et al. (2011)*, who had demonstrated oxidative stress causing NMJ growth. These structural NMJ changes are similar to those produced by neuronal overactivation (*Figure 2B*) and implicate ROS, specifically $H_2O_2$, as a signal for regulating activity-dependent structural plasticity at input and output terminals.

## DJ-1β acts as a ROS sensor in neurons

We then investigated how neurons might sense changes in activity-induced ROS levels. ROS are known to post-translationally modify many different proteins, principally on cysteine or methionine residues, including transcription factors, cytoskeletal proteins, cell adhesion molecules and phosphatases (for review see *Milton and Sweeney, 2012*). We identified DJ-1β, the fly ortholog of DJ-1 (PARK7), as a candidate ROS sensor. *DJ-1* codes for a highly conserved, ubiquitously expressed redox-sensitive protein, that protects against oxidative stress and regulates mitochondrial function (*Ariga et al., 2013*; *Nagakubo et al., 1997*). A mutant allele is also linked to a rare form of familial Parkinsonism (*Bonifati et al., 2003*). *DJ-1* null mutants are viable without any conspicuous defects and adult flies have been reported to have increased sensitivity to oxidative stress, for example as induced by exposure to paraquat or $H_2O_2$ (*Meulener et al., 2005*). We found that *DJ-1β* null mutant ($DJ-1β^{\Delta93}$) larvae develop normally, indistinguishable from wild type controls, and have structurally normal NMJs (*Figure 3A* and *Figure 3—figure supplement 1*). Interestingly, in larvae homozygous for the $DJ-1β^{\Delta93}$ null mutation, NMJs fail to respond to elevated ROS levels and do not generate additional boutons, as typical for controls when feeding 10 mM paraquat, which causes elevated $O_2^-$ release from mitochondrial complex 1, or 10 mM diethyl maleate (DEM), which inactivates the ROS scavenger glutathione (*Milton et al., 2011*) (*Figure 3A*). Similarly, loss of DJ-1β also significantly rescues NMJ bouton addition phenotypes induced by targeted expression of the ROS generator Duox in aCC and RP2 motoneurons, or their dTrpA1-mediated overactivation (*Figure 3A*). $DJ-1β^{\Delta93}$ mutant larvae also failed to produce the presynaptic bouton and active zone addition that result from raising larvae at 29°C (*Sigrist et al., 2003*) (*Figure 3—figure supplement 2*). The neuronal requirement for *DJ-1β* was further tested via rescue experiments where neuronal *UAS-DJ-1β* miss-expression in a *DJ-1β* null-mutant background proved sufficient to re-establish sensitivity to DEM (see *Figure 3—figure supplement 3*). Next, we cell-autonomously changed the ability of DJ-1β to act as a redox sensor. DJ-1 is known to form dimers (*Lin et al., 2012a*). We expressed in aCC and RP2 motoneurons a dominant-acting mutant form of DJ-1β that is non-oxidizable at the conserved cysteine 104, $UAS-DJ-1β^{C104A}$ (*Meulener et al., 2006*). Expression of $DJ-1β^{C104A}$ abrogated ROS-induced (following DEM feeding) as well as dTrpA1 activity-mediated NMJ structural adjustment, both with respect to bouton number (*Figure 3A*) and active zone number (*Figure 3B*). Looking at activity-dependent structural plasticity of the postsynaptic dendritic arbor, we found this is sensitive to *DJ-1β* levels; halving the copy number of *DJ-1β* (in $DJ-1β^{\Delta93}/+$ heterozygotes) was sufficient to significantly suppress dTrpA1-mediated changes to dendritic arbor size (*Figure 3C*).

In summary, these data show that DJ-1β is necessary for activity-induced structural changes, compatible with DJ-1β functioning as a sensor for ROS. DJ-1β appears to be required in motoneurons for increasing NMJ bouton and active zone numbers in response to mild overactivation regimes that lead to potentiation of transmission at the NMJ (raising larvae at 29°C) (*Sigrist et al., 2003*). At higher levels, as induced by dTrpA1-mediated overactivation, DJ-1β is also necessary in motoneurons, though under these conditions for decreasing active zone numbers at the NMJ and the size of postsynaptic dendritic arbors.

## PTEN and PI3K are downstream effectors of the DJ-1β ROS sensor

Next, we looked for downstream effector pathways responsible for implementing activity and ROS-dependent structural plasticity. DJ-1 is a known redox-regulated inhibitor of PTEN and as such disinhibits PI3Kinase signaling (*Kim et al., 2005*; *Kim et al., 2009b*). PI3Kinase was previously shown to regulate bouton and active zone number during NMJ development (*Jordán-Álvarez et al., 2012*;

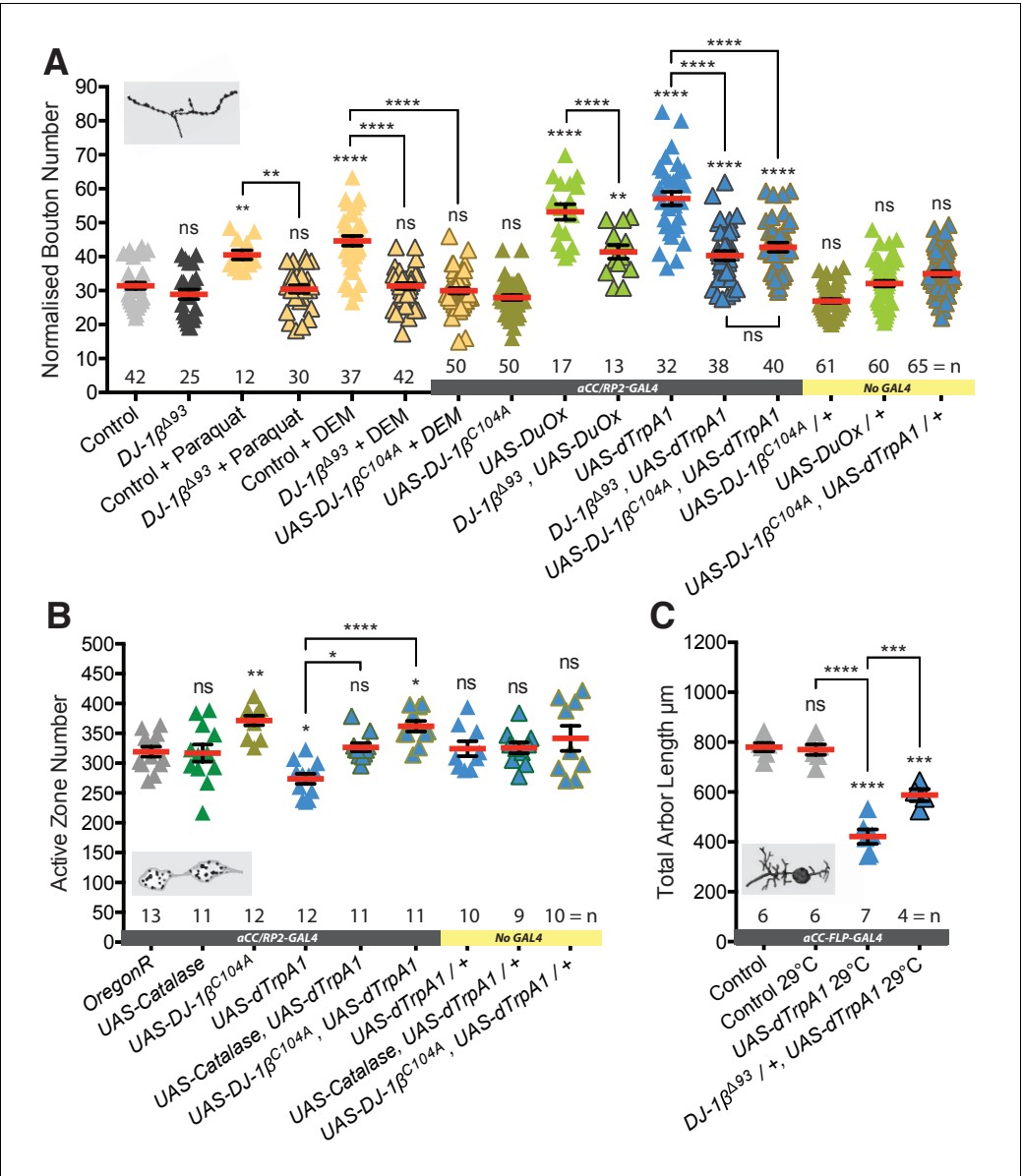

**Figure 3.** DJ-1β senses ROS and regulates activity-induced neural plasticity. (**A**) DJ-1β is required for ROS and neuronal-activity-induced NMJ elaboration (100 hr ALH). Larvae reared at 25°C. (**B**) Cell-specific expression of DJ-1β$^{C104A}$, non-oxidizable on conserved cysteine C104, prevents activity-induced reduction of active zone number. 'Control' is *aCC/RP2-GAL4* alone. Larvae were reared at 25°C. (**C**) Activity-generated ROS sensing is dose sensitive. Removal of one copy of *DJ-1β* (in *DJ-1β$^{Δ93}$ /+* heterozygotes) is sufficient to significantly rescue activity-induced reduction of total dendritic arbor length of motoneurons in 24 hr ALH larvae. 'Control' is *aCC-FLP-GAL4* alone, heterozygous, achieved by crossing the GAL4 driver to Oregon-R wild type flies.

DOI: https://doi.org/10.7554/eLife.39393.006

The following source data and figure supplements are available for figure 3:

**Source data 1.** Source data for *Figure 3*.

DOI: https://doi.org/10.7554/eLife.39393.010

**Figure supplement 1.** *DJ-1β$^{Δ93}$* mutant third instar larval NMJs are phenotypically normal.

DOI: https://doi.org/10.7554/eLife.39393.007

**Figure supplement 2.** *DJ-1β$^{Δ93}$* mutant third instar larval NMJs fail to elaborate bouton number or active zone number upon rearing at 29°C.

DOI: https://doi.org/10.7554/eLife.39393.008

**Figure supplement 3.** 10 mM DEM feeding induces increased bouton number at both muscle four and DA1.

*Figure 3 continued on next page*

Figure 3 continued

DOI: https://doi.org/10.7554/eLife.39393.009

*Martín-Peña et al., 2006*). To test if DJ-1β - PTEN interactions mediate ROS-dependent NMJ adjustments, we performed genetic interaction experiments in the context of DEM-induced ROS elevation (*Milton et al., 2011*) (*Figure 4A*). We used different DEM dosages to generate a dose-response curve and focused on changes in NMJ bouton number as a quantitative readout. We found that in controls bouton number increases linearly with exposure to increasing DEM concentrations, peaking at 10 mM DEM (*Figure 4A*). Removal of one copy of *DJ-1β* (*DJ-1β*$^{\Delta 93}$/+) was sufficient to suppress these DEM-induced increases in bouton number. Conversely, removing one copy of the PI3Kinase antagonist PTEN (*PTEN*$^{CO76}$/+) resulted in increased sensitivity to DEM, as indicated by the left-shifted dose response curve. Larvae made heterozygous mutant for both *DJ-1β* and *PTEN* (*PTEN*$^{CO76}$/+; *DJ-1β*$^{\Delta 93}$/+) were overall less sensitive to DEM than controls, though at higher concentrations displayed greater sensitivity to DEM than *DJ-1β*$^{\Delta 93}$/+ heterozygotes, as might be expected when lowering *PTEN* copy number. These genetic interactions support previous studies (*Kim et al., 2005*) and complement biochemical data that showed increased binding of DJ-1β to PTEN following oxidation by H$_2$O$_2$, thus effecting PTEN inhibition (*Kim et al., 2009b*).

To further test specificity, we manipulated PTEN and PI3Kinase activities in single cells. Targeted knock-down of endogenous PTEN in aCC and RP2 motoneurons sensitized these to ROS, exacerbating the DEM-induced bouton addition phenotype (*Figure 4B*). In contrast, over-expression of PTEN or mis-expression of a dominant negative form of PI3Kinase significantly reduced NMJ elaboration normally caused by DEM exposure or dTrpA1-mediated overactivation (*Figure 4B*). Together these genetic interactions suggest a working model whereby PTEN and PI3Kinase act downstream of DJ-1β; and neural activity-generated ROS, via oxidation of DJ-1β, leads to PTEN inhibition. This in turn

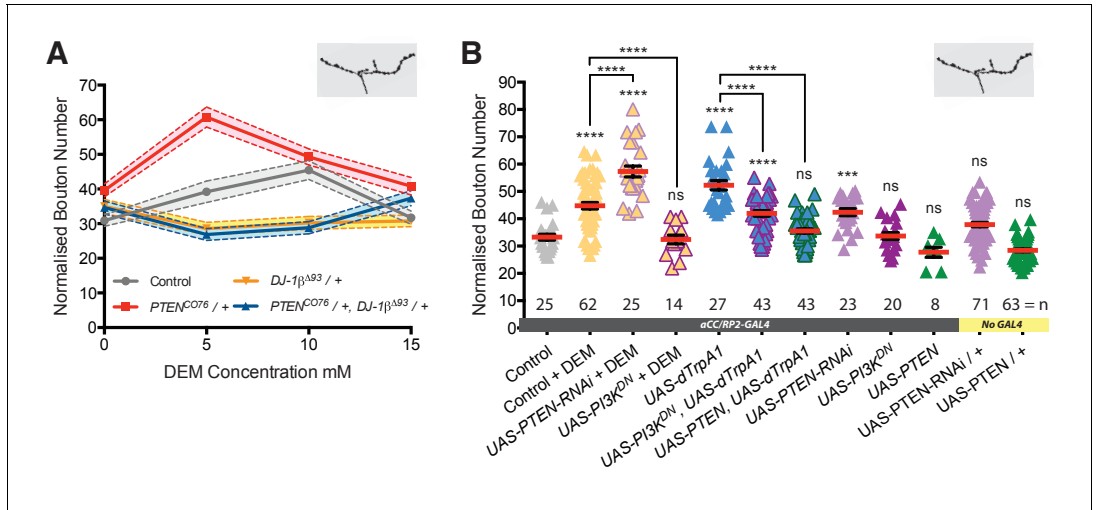

**Figure 4.** DJ-1β signals via PTEN and PI3Kinase to regulate ROS and activity-induced NMJ elaboration. (**A**) *DJ-1β* and *PTEN* genetically interact to regulate systemic ROS-induced NMJ elaboration. NMJ bouton number varies with ROS (DEM) levels (grey data). Removal of one copy of *PTEN* sensitizes (red) while heterozygosity for *DJ-1β* desensitizes NMJs to ROS levels (yellow), partially restored in double heterozygotes (blue). Dashed boundaries indicate 95% confidence intervals (n ≥ 38). (**B**) Systemic ROS and activity-induced NMJ structural adjustments require PTEN and PI3Kinase signaling. Over-expression of the PI3Kinase antagonist PTEN or a dominant negative PI3Kinase form abrogates activity-induced NMJ elaboration. 'Control' is aCC/RP2-GAL4 alone, heterozygous, achieved by crossing the GAL4 driver to Oregon-R wild type flies. Mean ± SEM, ANOVA, ns = not significant, **p<0.01, ***p<0.001, ****p<0.0001, n = replicate number.

DOI: https://doi.org/10.7554/eLife.39393.011

The following source data is available for figure 4:

**Source data 1.** Source data for *Figure 4*.

DOI: https://doi.org/10.7554/eLife.39393.012

facilitates a rise in PI3Kinase / PIP$_3$ signaling, which mediates at least part of the structural synaptic terminal plasticity by regulating bouton and active zone number at the NMJ (*Figure 7*).

## Activity-regulated ROS signaling is necessary for homeostatic adjustment of synaptic transmission at the NMJ

To complement our studies of structural plasticity at the NMJ, we carried out sharp electrode recordings from the target muscle co-innervated by the aCC and RP2 motoneurons, muscle dorsal acute 1 (DA1) (*Hoang and Chiba, 2001*; *Landgraf et al., 2003*; *Mauss et al., 2009*). In Oregon-R wild type larvae we recorded evoked excitatory junction potentials (eEJPs) of 18 ± 2 mV (*Figure 5A, D*) and miniature excitatory junction potentials (mEJPs) indicative of spontaneous vesicle fusion of 1 mV (*Figure 5B,D*). Therefore, on average each action potential triggers fusion of 18 vesicles at this NMJ (quantal content) (*Figure 5C*).

We investigated how these parameters of NMJ transmission might change following cell-specific dTrpA1-mediated overactivation of the aCC and RP2 motoneurons (at 25°C and 27°C). We then tested if these activity-driven changes in transmission required ROS signaling, by co-expressing the ROS scavenger Catalase or by abrogating ROS sensor function through co-expression of the dominant acting, non-oxidizable DJ-1β variant, DJ-1β$^{C104A}$. Transmission at NMJs in *Drosophila* is characteristically robust due to several homeostatic regulatory mechanisms working toward maintaining

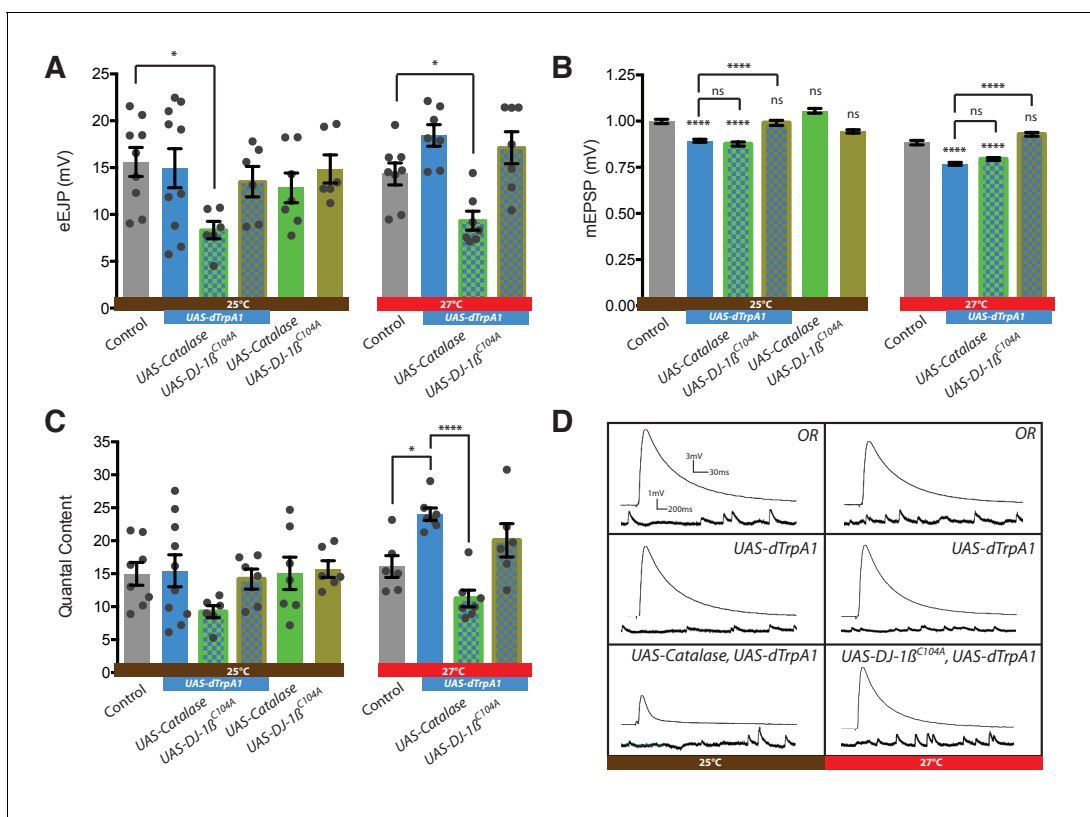

**Figure 5.** Presynaptic ROS regulate the maintenance of eEJP amplitude and DJ-1β function is required for the reduction of mEJP amplitude during overactivation. (A) Overactivated motoneurons (expressing dTrpA1 at 25°C, 27°C) show similar eEJP amplitude to control, despite reduced mEJP amplitude (see B). Catalase co-expression prevents this adaptation, significantly reducing eEJP amplitude, whereas DJ-1β appears not to be required. (B) Increased neuronal activity, mediated by dTrpA1 expression at 25°C or by elevating ambient temperature to 27°C, reduces mEJP amplitude. Co-expression of DJ-1β$^{C104A}$ rescues this effect. (C) Quantal content (eEJP amplitude/mEJP amplitude) shows no significant difference between groups due to high variance within some conditions. (D) Representative eEJP and mEJP traces. ANOVA, ns = not significant, *p<0.05, ****p<0.0001.
DOI: https://doi.org/10.7554/eLife.39393.013

The following source data is available for figure 5:

**Source data 1.** Source data for *Figure 5*.
DOI: https://doi.org/10.7554/eLife.39393.014

constancy of eEJPs (for reviews see *Frank, 2014*; *Frank et al., 2013*; *Harris and Littleton, 2015*). Following dTrpA1-mediated overactivation (rearing larvae at 25°C and 27°C) evoked NMJ transmission at the muscle DA1 NMJ remains intact and homeostatically balanced (*Figure 5A,D*). Recordings from controls and larvae with dTrpA1 expressing motoneurons reared at 27°C showed noticeably less variability of eEJP amplitude than those made from genetically identical siblings reared at 25°C, the dTrpA1 activation threshold, potentially reflecting variable efficacy in motoneuron manipulation (*Hamada et al., 2008*; *Pulver et al., 2009*). Overexpression of either Catalase or DJ-1β$^{C104A}$ has no effect on eEJP amplitude. However, co-expression of Catalase with dTrpA1-mediated overactivation leads to reduction of eEJP amplitude to 7–9 mV. This suggests that ROS are required for mechanisms that homeostatically maintain eEJP amplitude (*Figure 5A,D*).

In contrast, the amplitude of spontaneous vesicle fusion events (mEJP) adjusts inversely with progressively stronger overactivation of motoneurons. These reductions in mEJP amplitude are not specific to dTrpA1-mediated overactivation. Along with others (*Ueda and Wu, 2015*; *Yeates et al., 2017*) we also recorded similarly reduced mEJP amplitudes in wild type Oregon-R larvae that had been reared at an elevated temperature (27°C *versus* controls reared at 25°C) (*Figure 5B*). Co-expression of DJ-1β$^{C104A}$ abrogates this activity-dependent reduction of mEJP amplitude (*Figure 5B,D*), while Catalase co-expression has no effect on mEJP amplitude. Our analysis of quantal content (the mean number of vesicles releasing transmitter per action potential, calculated as the ratio of eEJP/mEJP amplitudes) showed that dTrpA1-mediated motoneuron over-activation at 27°C leads to significantly increased quantal content, and that this is brought back to control levels with co-expression of the $H_2O_2$ scavenger Catalase, as might be expected from the reduced eEJP amplitudes under such conditions.

In summary, evoked transmission at the NMJ is homeostatically maintained despite increased dTrpA1-mediated neuronal overactivation. The maintenance of eEJP amplitude requires ROS and is compromised when the $H_2O_2$ scavenger Catalase is expressed by the presynaptic motoneuron. Activity-regulated changes in mEJP amplitude also depend on ROS signaling, though this is not impacted by the over-expression of cytoplasmic Catalase; instead we found this aspect of synaptic transmission sensitive to oxidation of DJ-1β.

## Structural plasticity of synaptic terminals is required for homeostatic adjustment of locomotor behavior

We wondered what impact the observed activity-ROS-regulated structural adjustments might have on network output. To test this we used larval crawling speed as a quantifiable readout for a simple locomotor behavior. In agreement with previous studies (*Sigrist et al., 2003*; *Zhong and Wu, 2004*), we found that crawling speed increases upon acutely shifting mid-3$^{rd}$ instar larvae (72 hr ALH) to higher ambient temperatures (e.g., from 25°C to 29°C or 32°C) (blue data in *Figure 6A*). This is to be expected given that these animals have an innate preference for approximately 25°C (*Dillon et al., 2009*; *Hamada et al., 2008*; *Luo et al., 2017*; *Rosenzweig et al., 2005*). In contrast, following prolonged exposure to an elevated temperature, achieved by rearing larvae at 29°C or 32°C, larvae crawled at the same speed characteristic of controls reared at 25°C (average of 0.65–0.72 mm/second; grey horizontal dotted line in *Figure 6A*). This crawling speed adaptation is suggestive of a homeostatic adjustment of the locomotor network. Increased network drive might be counteracted by reduced neuronal excitability and/or synaptic input, thus allowing motor output to be returned to the default crawling speed; in which case one would expect greater adjustment in larvae reared at 32°C than 29°C. To reveal such adjustments we acutely shifted warmth-adapted animals to 25°C, which caused reduced crawling speed (green data in *Figure 6A*). Indeed, following this downward-shift 32°C-adjusted larvae crawled significantly slower than 29°C-adjusted animals, suggesting the degree of neuronal adjustment is proportional to the level of temperature-induced overactivation. These experiments suggest that during prolonged activity manipulations the larval locomotor network output homeostatically adjusts toward a default crawling speed.

Next, we wanted to test the role of the motoneurons in this form of network adjustment. The motoneurons integrate all pre-motor input within their dendritic arbors and produce the output of the motor network. We targeted expression of dTrpA1 to the motoneurons (and other glutamatergic cells, using *dVGlut-GAL4*) and reared these animals at 27°C, a temperature that robustly activates dTrpA1-expressing neurons (*Hamada et al., 2008*; *Pulver et al., 2009*). Upon acute removal of this

overstimulation (by shifting animals to 22°C where the dTrpA1 channel is closed) larval crawling speed reduced significantly relative to non-dTrpA1 expressing controls (*Figure 6B*). We then asked whether this ROS-mediated structural plasticity of synaptic terminal growth was required for homeostatic adjustment of larval crawling speed, which ensues after prolonged overactivation. We manipulated the ability of neurons to sense increases in ROS levels by targeting expression of the dominant acting non-oxidizable DJ-1β$^{C104A}$ variant to motoneurons. We then tested the behavior of these animals for adjustment in response to chronic temperature-induced elevation of motor network activity. Our previous set of experiments showed that expression of non-oxidizable DJ-1β$^{C104A}$ in motoneurons prevents structural adjustment of bouton number and the decrease in active zone number normally caused by overactivation (see *Figure 3A and B*). Expression of non-oxidizable DJ-1β$^{C104A}$ in motoneurons per se did not alter larval crawling speed at the control temperature of 25°C. However, when rearing these larvae at 32°C, which is associated with elevated motor network activation, unlike controls they failed to homeostatically adjust toward the default crawling speed (*Figure 6C*). Consequently, such larvae reared at elevated temperatures (29°C or 32°C) also responded less strongly than controls to acute temperature downshifts (*Figure 6C*). These data suggest that activity-induced structural plasticity, implemented through the ROS sensor DJ-1β, is necessary for activity-directed homeostatic adjustment of larval locomotor behavior.

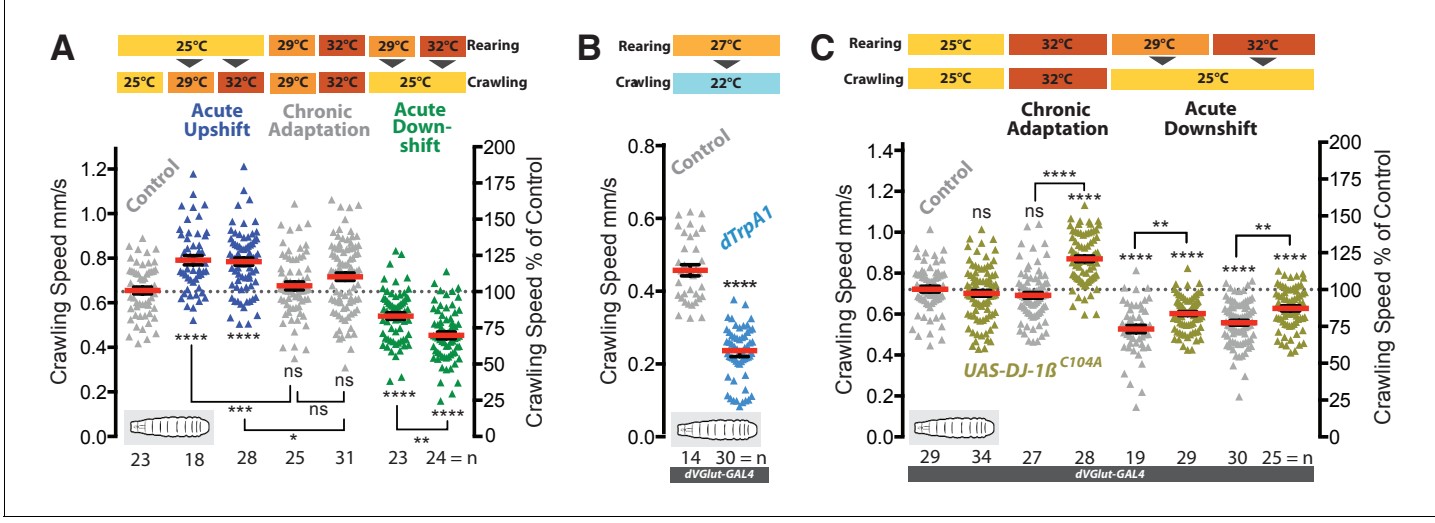

**Figure 6.** Adaptive behavioral plasticity in response to chronic locomotor overactivation. (**A**) Larval motor network activity, assayed by crawling speed 72 hr after larval hatching (AHL), increases in response to acute temperature upshifts (blue) in wild type larvae. In contrast, with prolonged exposure (grey) to elevated temperatures (29°C or 32°C) the motor network adapts homeostatically generating the same crawling speed as 25°C reared controls. This adaptation is further revealed by acute temperature downshifts (green). Each data point represents crawling speed from an individual uninterrupted continuous forward crawl, n = specimen replicate number, up to three crawls assayed for each larva. Genotype = OregonR. (**B**) Prolonged overactivation targeted to motoneurons (*dVGlut-GAL4; UAS-dTrpA1*) also leads to adaptation with reduced crawling speed (dTRPA1 channels open at 27°C, closed at 22°C). Mean ±SEM, control is *dVGlut-GAL4 / +*. ****p<0.0001 students t test, n = replicate number. (**C**) Larvae with expression of *UAS-DJ-1β$^{C104A}$* targeted to motoneurons (*dVGlut-GAL4*) are unable to adapt motor network output (crawling speed) to elevated rearing temperatures. Control is *dVGlut-GAL4* alone, in heterozygous condition. Each data point represents crawling speed from an individual uninterrupted continuous forward crawl, n = specimen replicate number, up to three crawls assayed for each larva. Mean ± SEM, ANOVA, ns = not significant, *p<0.05, **p<0.01, ***p<0.001, ****p<0.0001.
DOI: https://doi.org/10.7554/eLife.39393.015

The following source data is available for figure 6:

**Source data 1.** Source data for *Figure 6*.
DOI: https://doi.org/10.7554/eLife.39393.016

## Discussion

### ROS signaling is required for neuronal activity-dependent structural plasticity

Building on previous work that had shown oxidative stress as inducing NMJ growth (*Milton et al., 2011*), we have in this study identified ROS as obligatory signals for activity-regulated structural plasticity (*Figure 7*). We further show that ROS are also sufficient to bring about structural changes at synaptic terminals that largely mimic those induced by neuronal overactivation (*Figure 2B,C*). A mitochondrially targeted ROS reporter (*Albrecht et al., 2011*; *Gutscher et al., 2009*) suggests a positive correlation between levels of neuronal activity and ROS generated in mitochondria, potentially as a byproduct of increased ATP metabolism or triggered by mitochondrial calcium influx (*Peng and Jou, 2010*) (*Figure 2A*). Although we did not specifically investigate the nature of the active ROS in this context, three lines of evidence suggest that $H_2O_2$, generated by the dismutation of $O_2^-$, is the principal signaling species. First, under conditions of neuronal overactivation (but not control levels of activity) over-expression of the $O_2^-$ to $H_2O_2$ converting enzyme SOD2 potentiated structural plasticity phenotypes (*Figure 2B*). Second, over-expression of the $H_2O_2$ scavenger Catalase efficiently counter-acts all activity-induced changes we have quantified, at both postsynaptic dendritic and presynaptic NMJ terminals. Third, over-expression of the $H_2O_2$ generator Duox in motoneurons is sufficient to induce NMJ bouton phenotypes that mimic overactivation (*Figure 3A*). In addition to mitochondria, other sources of ROS include several oxidases, notably NADPH oxidases. These have been implicated during nervous system development in the regulation of axon growth and synaptic plasticity (*Kishida et al., 2006*; *Munnamalai and Suter, 2009*; *Munnamalai et al., 2014*; *Olguín-Albuerne and Morán, 2015*; *Serrano et al., 2003*; *Tejada-Simon et al., 2005*; *Wilson et al., 2016*; *Wilson et al., 2015*). NADPH oxidases can be regulated by NMDA receptor stimulation (*Brennan et al., 2009*) and activity-associated pathways, including calcium, Protein kinases C and A and calcium/calmodulin-dependent kinase II (CamKII) (*Bánfi et al., 2004*; *Massaad and Klann, 2011*; *Pandey et al., 2011*; *Sorce et al., 2017*; *Tirone and Cox, 2007*). The precise sources of activity-regulated ROS, potentially for distinct roles in plasticity, will be interesting to investigate.

### ROS as gatekeepers of activity-dependent synaptic structural plasticity

We demonstrated that ROS are necessary for activity-dependent structural plasticity of *Drosophila* motoneurons, at both their postsynaptic dendrites in the CNS and presynaptic NMJs in the periphery (*Figure 7*). The mechanisms by which ROS intersect with other known plasticity pathways now need to be investigated. Among well documented signaling pathways regulating synaptic plasticity, are Wnts (*Budnik and Salinas, 2011*), BMPs (*Bayat et al., 2011*; *Berke et al., 2013*), PKA, CREB and the immediate early gene transcription factor AP-1 (*Cho et al., 2015*; *Davis, 2006*; *Davis and Müller, 2015*; *Davis et al., 1998*; *Davis et al., 1996*; *Kim et al., 2009a*; *Koles and Budnik, 2012*; *Osses and Henríquez, 2014*; *Sanyal et al., 2003*; *Sanyal et al., 2002*; *Sulkowski et al., 2014*; *Walker et al., 2013*). ROS signaling could be synergistic with other neuronal plasticity pathways, potentially integrating metabolic feedback. Indeed, ROS modulate BMP signaling in cultured sympathetic neurons (*Chandrasekaran et al., 2015*) and Wnt pathways in non-neuronal cells (*Funato et al., 2006*; *Love et al., 2013*; *Rharass et al., 2014*). Biochemically, ROS are well known regulators of kinase pathways via oxidation-mediated inhibition of phosphatases (*Finkel and Holbrook, 2000*; *Tonks, 2006*). Redox modifications also regulate the activity of the immediate early genes Jun and Fos, which are required for LTP in vertebrates, and in *Drosophila* for activity-dependent plasticity of motoneurons, both at the NMJ and central dendrites (*Hartwig et al., 2008*; *Jindra et al., 2004*; *Loebrich and Nedivi, 2009*; *Milton and Sweeney, 2012*; *Milton et al., 2011*; *Sanyal et al., 2002*). We therefore hypothesize that ROS may provide neuronal activity-regulated modulation of multiple canonical synaptic plasticity pathways.

### Structural plasticity and its coordination between pre- and postsynaptic terminals

We focused on three aspects of synaptic terminal plasticity: dendritic arbor size in the CNS, and bouton and active zone numbers at the NMJ. We used these as phenotypic indicators for activity-

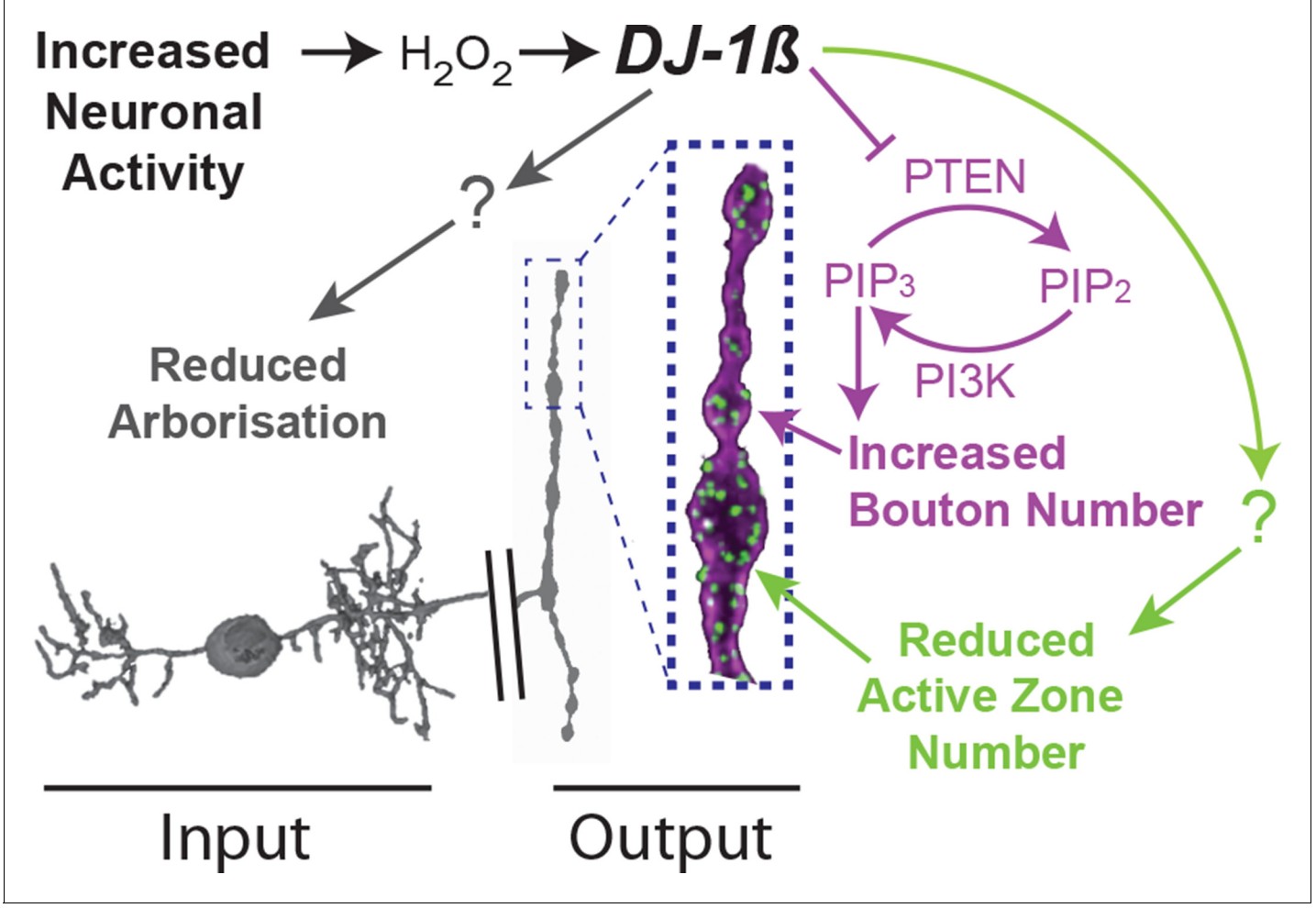

**Figure 7.** Model summary. DJ-1β is a redox signaling hub that coordinates structural synaptic plasticity at motoneuron synaptic input and output terminals. Activity-induced ROS oxidize DJ-1β, leading to PTEN inhibition and thus to a gain in PI3Kinase signaling, which regulates activity-induced NMJ elaboration of boutons and active zones. At higher activity/ROS thresholds additional, yet to be defined, pathways downstream of DJ-1β are activated, implementing adaptive reductions of active zones at the NMJ and dendritic arbor length in the CNS.

DOI: https://doi.org/10.7554/eLife.39393.017

regulated changes. By working with identified motoneurons we could observe adaptations across the entire neuron, relating adjustments of postsynaptic dendritic input terminals in the CNS to changes of the presynaptic output terminals at the NMJ in the periphery (*Figure 7*). For the aCC motoneuron, the degree of neuronal overactivation correlates with changes in synaptic terminal growth: notably reductions of dendritic arbor size centrally and of active zones at the NMJ. Interestingly, presynaptic active zone numbers did not show a linear response profile. Within a certain range low-level activity increases lead to more active zones, associated with potentiation (as previously shown by *Sigrist et al., 2003*); however, with stronger overactivation active zone number decrease (*Figure 1D,E*). Reduction of active zones, as we observed at the NMJ, and of Brp levels by increased activation was previously also reported in photoreceptor terminals of the *Drosophila* adult visual system (*Sugie et al., 2015*). At a finer level of resolution it will be interesting to determine how these activity-ROS-mediated structural changes might change active zone cytomatrix composition, which can impact on transmission properties, such as vesicle release probability (*Davydova et al., 2014*; *Lazarevic et al., 2011*; *Matz et al., 2010*; *Peled et al., 2014*; *Weyhersmüller et al., 2011*)

We previously found that in these motoneurons dendritic length correlates with the number of input synapses and with synaptic drive (*Zwart et al., 2013*). Therefore, we tentatively interpret the negative correlation between the degree of overactivation and the reduction in central dendritic

arbors as compensatory. In agreement, we found that blockade of activity-induced structural adjustment targeted to the motoneurons prevents behavioral adaptation normally seen after prolonged overactivation (*Figure 6*). Less clear is if and how overactivation-induced structural changes at the NMJ might be adaptive. Unlike many central synapses that facilitate graded analogue computation, the NMJ is a highly specialized synapse with a large safety factor and intricate mechanisms that ensure constancy of evoked transmission in essentially digital format (*Frank et al., 2013*; *Marrus and DiAntonio, 2005*). Rearing larvae at 29°C (which acutely increases motor activity) leads to more active zones at the NMJ and potentiated transmission, yet these larvae crawl at the same default speed as other larvae reared at 25°C (control) or 32°C with reduced numbers of active zones (compare *Figure 1E* with *Figure 6B*). This suggests that at least with regard to regulating crawling speed, plasticity mechanisms probably operate at the network level, rather than transmission properties of the NMJ. Indeed, our recordings of transmission at the NMJ, and those reported by others, show homeostatic maintenance of eEJP amplitude irrespective of changes in bouton and active zone number (*Figure 5*) (*Campbell and Ganetzky, 2012*). Though in this study we focused on anatomical changes, we expect these structural adjustments to be linked to, and probably preceded by compensatory changes in neuronal excitability that have been documented (*Baines et al., 2001*; *Davis, 2006*; *Davis et al., 1996*; *Davis et al., 1998*; *Driscoll et al., 2013*; *Frank et al., 2006*; *Frank et al., 2009*; *Gaviño et al., 2015*; *Giachello and Baines, 2017*; *Lin et al., 2012b*; *Mee et al., 2004*; *Müller and Davis, 2012*; *Wang et al., 2014*; *Younger et al., 2013*; *O'Leary et al., 2013*; *Prinz, 2006*; *Prinz et al., 2004*).

Our observations of activity-regulated adjustments of both dendritic arbor size and NMJ structure give the impression of processes coordinated across the entire neuron. If this was the case, it could be mediated by transcriptional changes, potentially via immediate early genes (AP-1), which are involved in activity and ROS-induced structural changes at the NMJ (*Milton et al., 2011*; *Sanyal et al., 2002*) and motoneuron dendrites (*Hartwig et al., 2008*).

## Identification of DJ-1β as a neuronal ROS sensor

We discovered that in neurons the highly conserved protein DJ-1β is critical for both structural and physiological changes in response to activity-generated ROS (*Figure 3* and *Figure 5*). In neurons DJ-1β might act as a redox sensor for activity-generated ROS. In agreement with this idea, DJ-1β has been shown to be oxidized by $H_2O_2$ at the conserved cysteine residue C106 (C104 in *Drosophila*) (*Lin et al., 2012a*; *Meulener et al., 2006*). Oxidation of DJ-1 leads to changes in DJ-1 function, including translocation from the cytoplasm to the mitochondrial matrix, aiding protection against oxidative damage (*Blackinton et al., 2009*; *Canet-Avilés et al., 2004*; *Waak et al., 2009*) and maintenance of ATP levels (*Calì et al., 2015*). We found that the ability of motoneurons to respond to increased activation is potently sensitive to DJ-1β dosage. It is also blocked by expression of mutant DJ-1β$^{C104A}$ that is non-oxidisable on the conserved Cys104 (*Hao et al., 2010*; *Meulener et al., 2006*). These observations suggest that DJ-1β is critical to ROS sensing in neurons. They also predict that cell type-specific DJ-1β levels, and associated DJ-1β reducing mechanisms, could contribute to setting cell type-specific sensitivity thresholds to neuronal activity.

## DJ-1βdownstream pathways implement activity-regulated plasticity

Our data suggest that DJ-1β could potentially be part of a signaling hub. At the NMJ, this might mediate plasticity across a range, from the addition of active zones associated with potentiation to, following stronger overactivation, the reduction of active zones. We identified disinhibition of PI3Kinase signaling as one DJ-1β downstream pathway (*Figure 4*) (*Kim et al., 2005*; *Kim et al., 2009b*), a well-studied intermediate in metabolic pathways and a known regulator of synaptic terminal growth, including active zone addition (*Jordán-Álvarez et al., 2012*; *Martín-Peña et al., 2006*). However, with stronger overactivation DJ-1β might engage additional downstream effectors that reduce active zone addition or maintenance, potentially promoting active zone disassembly. While at the presynaptic NMJ PI3Kinase disinhibition explains activity-regulated changes in bouton addition, different DJ-1β effectors likely operate in the somato-dendritic compartment, which responds to overactivation with reduced growth and possibly pruning (*Brierley et al., 2009*). Thus, sub-cellular compartmentalization of the activity-ROS-DJ-1β signaling axis could produce distinct plasticity responses in pre- *versus* postsynaptic terminals.

# Homeostatic maintenance of synaptic transmission requires presynaptic ROS signaling

Previous studies demonstrated a requirement for ROS for LTP (*Huddleston et al., 2008*; *Kamsler and Segal, 2003a*; *Kamsler and Segal, 2003b*; *Klann, 1998*; *Knapp and Klann, 2002*; *Lee et al., 2010*) and found learning defects in animal models with reduced NADPH oxidase activity (*Kishida et al., 2006*), suggesting that synaptic ROS signaling might be a conserved feature of communication in the nervous system. Our sharp electrode recordings from muscle DA1 revealed three interesting aspects. First, that changing ROS signaling in the presynaptic motoneuron under normal activity conditions does not obviously impact on NMJ transmission. Second, quenching of presynaptic ROS by expression of Catalase under overactivation conditions led to a significant decrease in eEJP amplitude and concomitantly reduced quantal content (*Figure 5A,C*). This shows that upon chronic neuronal overactivation ROS signaling is critically required in the presynaptic motoneuron for maintaining eEJP amplitude by increasing vesicle release at the NMJ. This could be achieved by increasing vesicle release probability, which would counteract the reduction in active zone number following a period of neuronal overactivation. In this context it is interesting that components of the presynaptic release machinery, including SNAP25, are thought to be directly modulated by ROS (*Giniatullin et al., 2006*), while others, such as Complexin, might be indirectly affected, for example via ROS-mediated inhibition of phosphatases leading disinhibition of kinase activity (*Cho et al., 2015*). Third, we found that overactivation of motoneurons leads to reduced mEJP amplitude, also recently reported by others (*Yeates et al., 2017*). Curiously, mEJP amplitude, unlike eEJP amplitude, is regulated by DJ-1β, but is not impacted on by artificially increased cytoplasmic levels of the $H_2O_2$ scavenger Catalase. How it is that under conditions of neuronal overactivation eEJP and mEJP amplitudes are differentially sensitive to cytoplasmic Catalase *versus* DJ-1β oxidation is unclear, though it marks these two processes as distinct. One possibility is that cytoplasmic Catalase changes the local redox status, which could directly affect the properties of the presynaptic active zone cytomatrix. In contrast, mEJP amplitude regulation might be indirect and cell non-autonomous, via modulation of glutamate receptors in the postsynaptic target muscle (*Davis et al., 1998*).

Thus, several distinct ROS responsive pathways appear to operate at the NMJ. Structural adjustments in terms of synaptic terminal growth and synapse number are mediated by mechanisms sensitive to DJ-1β oxidation, potentially regulated via local reducing systems, including Catalase. In addition and distinct from these structural changes, at least in part, are the ROS-regulated adjustments in synaptic transmission that show different ROS sensitivities, one maintaining quantal content of evoked transmission while the other reduces mEJP amplitude when neuronal activity goes up (*Figure 5B*). It is conceivable that spatially distinct sources of ROS, for example mitochondria *versus* membrane localized NADPH oxidases, with different temporal dynamics could potentially mediate such differences in ROS sensitivities at the NMJ.

# Homeostatic adjustment of larval crawling speed depends on redox modification of DJ-1β

Our experiments exploring the potential behavioral relevance of activity-regulated structural plasticity demonstrated that network drive is regulated by ambient temperature. Acute elevation in ambient temperature produces faster crawling, while acute temperature reduction has the opposite effect. In contrast, with chronic temperature manipulations, larval crawling returns to its default speed (approx. 0.65–0.72 mm/sec) (*Figure 6A*). This adaptation to chronic manipulations might overall be energetically more favorable. It also allows larvae to retain a dynamic range of responses to relative changes in ambient temperature (i.e. speeding up or slowing down).

Where in the locomotor network these adjustments take place remains to be worked out. It is reasonable to assume that proprioceptive sensory neurons, and potentially also central recurrent connections, provide feedback information that facilitates homeostatic adjustment of network output. Our manipulations of the glutamatergic motoneurons show these cells are clearly important. For example, cell type-specific overactivation of the glutamatergic motoneurons (via dTrpA1) on the one hand, and blockade of activity-induced structural adjustment (by mis-expression of non-oxidizable DJ-1β$^{C104A}$) on the other demonstrated that ROS-DJ-1β-mediated processes that we showed important for structural adjustment are also required for implementing homeostatic tuning of locomotor network output (*Figure 6B,C*). The capacity of motoneurons as important elements in shaping

motor network output, might be explicable in that these neurons constitute the final integrators on which all pre-motor inputs converge (*Fushiki et al., 2016*; *Itakura et al., 2015*; *Kohsaka et al., 2014*; *Schneider-Mizell et al., 2016*; *Zwart et al., 2016*).

In conclusion, here we identified ROS in neurons as novel signals that are critical for activity-induced structural plasticity (*Figure 7*). ROS levels regulated by neuronal activity have the potential for operating as metabolic feedback signals. We further identified the conserved redox-sensitive protein DJ-1β as important to neuronal ROS sensing, and the PTEN/PI3Kinase synaptic growth pathway as a downstream effector pathway for NMJ growth in response to neuronal overactivation. These findings suggest that in the nervous system ROS operate as feedback signals that inform cells about their activity levels. The observation that ROS are important signals for homeostatic processes explains why ROS buffering is comparatively low in neurons (*Bell et al., 2015*). This view also shines a new light on the potential impact of ROS dysregulation with age or under neurodegenerative conditions, potentially interfering with neuronal adaptive adjustments and thereby contributing to network malfunction and synapse loss.

## Materials and methods

### Electrophysiology

Late wandering third instar larvae were fillet dissected in standard HL3 buffer (adapted from *Stewart et al., 1994*), 70 mM NaCl, 5 mM KCl, 20 mM $MgCl_2$, 10 mM $NaHCO_3$, 115 mM Sucrose, 5 mM HEPES, 1.5 mM CaCl, pH 7.25) ventral surface down with a lateral incision in order preserve both the ventral and dorsal midlines. Suction (GC150F-10 Harvard Apparatus) and sharp (GC100F-10 Harvard Apparatus) electrodes were pulled using a P-97 pipette puller (Sutter Instrument Company). Sharp electrode muscle impalement (DA1 muscle, at the dorsal midline) and inter-segmental nerve suction (ventral midline) were performed using an Olympus BX50WI compound microscope with 10X air (Olympus 10x/0.25 N.A.) and 20X dipping (Olympus UMPlanFL 20x/0.5 N.A.) objective lenses. Recordings were made at 21°C in HL3 using an Axopatch-1D amplifier (Axon Instruments), a 1322A DigiData (Axon Instruments), a DS2A-MkII Constant Voltage Isolated Stimulator (Digitimer Ltd.) and pCLAMP 10.4 acquisition software (Molecular Devices). mEJP (2–3 min) and eEJP (3 rounds of 20 stimulations, 5V) recordings were made in current-clamp mode from muscle cells with an input resistance above 8MOhm and a stable resting membrane potential between −40 mV and −70 mV. Analysis of eEJPs was performed using Clampfit10.6 (Molecular Devices) and mEJPs using Mini-Analysis6.0.7 (Synaptosoft).

### Fly strains and husbandry

Wild-type and transgenic strains were maintained on standard yeast–agar–cornmeal medium at 25°C. The following fly strains were used: *OregonR* and *PTEN^{CO76}* (Bloomington Stock Center, Indiana University), *UAS-dTrpA1* (*Hamada et al., 2008*), *UAS-SOD2* (*Missirlis et al., 2003*), *UAS-Catalase* (*Missirlis et al., 2001*), *UAS-Duox* (*Ha et al., 2005*), *DJ-1β^{Δ93}* (*Meulener et al., 2005*), *UAS-DJ-1β^{C104A}* (*Meulener et al., 2006*), UAS-RNAi lines targeting *SOD1*, *SOD2*, *Catalase* and *PTEN* (KK collection, Vienna Drosophila Resource Centre) (*Dietzl et al., 2007*), *UAS-PI3K^{DN}* (*Leevers et al., 1996*), *UAS-PTEN* (*Gao et al., 2000*). The following two GAL4 expression lines were used to target GAL4 to the aCC and RP2 motoneurons: *aCC-FLP-GAL4* (*eveRN2-Flippase, UAS-myr::mRFP1, UAS-Flp, tubulin84B-FRT-CD2-FRT-GAL4*) (*Roy et al., 2007*) expresses GAL4 stochastically in single aCC and RP2 motoneurons allowing the imaging of aCC neurons in isolation, as required for dendritic arbor resolution and reconstruction. *aCC/RP2-GAL4* (*eveRN2-GAL4* [*Fujioka et al., 2003*], *UAS-myr-mRFP1, UAS-Flp, tubulin84B-FRT-CD2-FRT-GAL4; RRaGAL4, 20xUAS-6XmCherry::HA* [*Shearin et al., 2014*]) was used for NMJ analysis as it expresses GAL4 in every aCC and RP2 motoneuron. *eveRN2-GAL4* expression is restricted to the embryo and FLPase-gated *tubulin84B-FRT-CD2-FRT-GAL4* maintains GAL4 expression at high levels during larval stages.

### Dissection and immunocytochemistry

#### 1st instar Ventral Nerve Cord (VNC)

Flies were allowed to lay eggs on apple juice-based agar medium for 24 hr at 25°C. Embryos were dechorionated using bleach (3.5 min room temperature) then incubated (25°C) in pre-warmed

Sorensen's saline (pH 7.2, 0.075 M) whilst adhered to a petri dish. Hatched larvae (floating) were recovered hourly and transferred to pre-warmed apple-juice agar plates supplemented with yeast paste. Larvae were allowed to develop for a further 24 hr (24 hr after larval hatching, ALH) at 25℃, 27℃ or 29℃ prior to dissection in Sorensen's saline. A fine hypodermic needle (30 1/2 G; Micro-lance) was used as a scalpel to cut off the anterior end of each larva, allowing gut, fat body, and tra-chea to be removed. The ventral nerve chord and brain lobes, extruded with viscera upon decapitation, were dissected out and transferred to a cover glass coated with poly-L-lysine (Sigma-Aldrich), positioned dorsal side up in Sorensen's saline. A clean cover glass was placed on top of the preparation, with strips of double-sided sticky tape as spacers positioned along the edges.

## Wandering third Instar

Flies were allowed to lay eggs on apple-juice agar based medium overnight at 25℃, larvae were then incubated at 25℃ or 27℃ until the late wandering third instar stage. Larvae were reared on yeast paste colored with Bromophenol Blue Sodium Salt (Sigma-Aldrich) to allow visualization of gut-clearance, an indicator of the late wandering third instar stage. For diethyl maleate (DEM) (Sigma-Aldrich) and paraquat (Sigma-Aldrich) feeding, yeast paste was made using a 5 mM – 15 mM aqueous solution. Larvae were 'fillet' dissected in Sorensen's saline and fixed for 15 min at room temperature in 4% formaldehyde (in Sorenson's saline). Specimens were then washed and stained in Sorensen's saline containing 0.3% Triton X-100 (Sigma-Aldrich) using the following primary/second-ary antibodies; Goat-anti-HRP Alexa Fluor 594 (1:400) (Jackson ImmunoResearch Cat. No. 123-585-021), Rabbit-anti-dsRED (1:1200) (ClonTech Cat. No. 632496), Donkey-anti-Rabbit CF568 (1:1200) (Biotium Cat. No. 20098) incubated overnight at 4℃ or 2 hr at room temperature. Specimens were mounted in EverBrite mounting medium (Biotium).

## Image acquisition and analysis

### 1st instar Ventral Nerve Cord (VNC)

Ventral nerve cords were pre-screened for fluorescently labeled, isolated, aCC motoneurons using a Zeiss Axiophot compound epifluorescence microscope and a Zeiss Plan-Neofluar 40x/0.75 N.A. objective lens. Suitable VNCs were imaged immediately with a Yokagawa CSU-22 spinning disk con-focal field scanner mounted on an Olympus BX51WI microscope, using a 60×/1.2 N.A. Olympus water immersion objective. Images were acquired with a voxel size of 0.2 × 0.2 × 0.3 μm using a QuantEM cooled EMCCD camera (Photometrics), operated via MetaMorph software (Molecular Devices). Dendritic trees were digitally reconstructed using Amira Resolve RT 4.1 (Visualization Scien-ces Group and Zuse Institute), supplemented with a 3D reconstruction algorithm (*Evers et al., 2005*; *Schmitt et al., 2004*), and images were processed using Amira and ImageJ (National Institutes of Health).

### Wandering third Instar

Dissected specimens were imaged using a Leica SP5 point-scanning confocal, and a 63x/1.3 N.A. (Leica) glycerol immersion objective lens and LAS AF (Leica Application Suite Advanced Fluores-cence) software. Confocal images were processed using ImageJ and Photoshop (Adobe). Bouton number of the NMJ on muscle DA1 (*Adler et al., 1999*) from segments A3-A5 was determined by counting every distinct spherical varicosity along the NMJ branch. DA1 muscles were imaged using a Zeiss Axiophot compound microscope and a Zeiss Plan-Neofluar 10x/0.3 N.A. objective lens. Mus-cle surface area (MSA) was determined by multiplying muscle length by width using ImageJ. In order to correct for subtle differences in animal size (typically 5–10%) bouton number normalization was performed using the following calculation: (mean control MSA/mean experimental MSA) x test bou-ton number = normalized experimental bouton number.

### Ratiometric ROS reporter

*aCC/RP2-Gal4* was used to drive the expression of *UAS-mito-roGFP2-Orp1* (*Albrecht et al., 2011*; *Gutscher et al., 2009*) in all aCC and RP2 motoneurons. Wandering third instar larvae were fillet dis-sected in PBS-NEM (137 mM NaCl, 2.7 mM KCl, 10 mM $Na_2HPO_4$, 1.8 mM $KH_2PO_4$, 20 mM N-ethyl-maleimide (NEM), pH 7.4). Larval fillet preparations were incubated for 5 min in PBS-NEM then fixed for 8 min in 4% formaldehyde (in PBS-NEM). Specimens were washed three times in PBS-NEM and

then equilibrated in 70% glycerol. Specimens were mounted in glycerol and imaged the same day. Imaging was performed on a Leica SP5 point-scanning confocal, using a 63x/1.3 N.A. (Leica) glycerol immersion objective lens. The reporter was excited sequentially at 405 nm and 488 nm (*Albrecht et al., 2011*) with emission detected at 500–535 nm. 16-bit images were acquired using Leica LAS AF software and processed using ImageJ. Z-stack images were maximally projected and converted to 32-bit. To remove fringing artefacts around bouton edges 488 nm images were thresholded using the 'Intermodes' algorithm with values below threshold set to 'not a number', and ratio images were created by dividing the 405 nm image by the 488 nm image pixel by pixel (*Albrecht et al., 2011*). Regions of Interest were taken on the ratio image spanning the entire NMJ and the mean value obtained from each NMJ was used for statistical analysis.

### Transmission electron microscopy

Third instar wandering larvae were fillet dissected in PBS and fixed overnight in 0.1M $NaPO_4$, pH 7.4, 1% glutaraldehyde, and 4% formaldehyde, pH 7.3. Fixed specimens were washed 3 × in 0.1M $NaPO_4$ before incubation in $OsO_4$ (1% in 0.1M $NaPO_4$; 2 hr). Preparations were washed 3 × in distilled water, incubated in 1% uranyl acetate, then washed again (3 × distilled water) and dehydrated through a graded ethanol series: 20% increments starting at 30% followed by two 100% changes and then 2 × 100% propylene oxide. Specimens were incubated in a graded series of epon araldite resin (in propylene oxide): 25% increments culminating in 3 × 100% changes. Individual muscles were then dissected and transferred into embedding molds, followed by polymerization at 60°C for 48 hr. Resin mounted specimens were sectioned (60–70 nm) using glass knives upon a microtome (Ultracut UCT; Leica). Sections were placed onto grids, incubated in uranyl acetate (50% in ethanol), washed in distilled water and incubated in lead citrate. Sections were imaged using a transmission electron microscope (TECNAI 12 $G^2$; FEI) with a camera (Soft Imaging Solutions MegaView; Olympus) and Tecnai user interface v2.1.8 and analySIS v3.2 (Soft Imaging Systems).

### Behavior – larval crawling analysis

To record larval crawling, mid-3$^{rd}$ instar larvae (72 hr ALH) were briefly rinsed in water to remove any food and yeast residues, then up to 12 larvae were placed into a 24 cm x 24 cm arena of 0.8% agar in water, poured to 5 mm thickness. Crawling behavior was recorded in a temperature and humidity controlled incubator at temperatures ranging from 25–32°C, as indicated for each experiment. Larvae were allowed to acclimatise for 5 min, then recorded for 15 min under infrared LED illumination (intensity from 14.33 nW/mm$^2$ in the edge to 9.12 nW/mm$^2$ in the center), using frustrated total internal reflection using a modified FIM tracker (*Risse et al., 2013*) https://www.uni-muenster.de/PRIA/en/FIM/index.html. Larvae were recorded with a Basler acA2040-180km CMOS camera using Pylon and StreamPix software, mounted with a 16 mm KOWA IJM3sHC.SW VIS-NIR Lens. Images were acquired at five frames per second. For each larvae, average crawling speed was calculated from long, uninterrupted forward crawls identified manually using FIMTrack. The 15 min recording period was partitioned into 5 min sections with each larvae being assayed once within each section, allowing each specimen to be sampled a maximum of 3 times. We observed no change in average crawling speed within the duration of the 15 min recording.

### Data analysis

All data handling was performed using Prism software (GraphPad). NMJ bouton number and ratiometric ROS reporter data were tested for normal/Gaussian distribution using the D'Agostino-Pearson omnibus normality test. Due to a lower replicate number, dendritic arbor reconstruction data were tested for normality using the Kolmogorov-Smirnov with Dallal-Wilkinson-Lilliefor P value test. Normal distribution was thus confirmed for all data presented, which were compared using one-way analysis of variance (ANOVA), with Tukey's multiple comparisons test where $*p<0.05$, $**p<0.01$, $***p<0.001$, $****p<0.0001$.

## Acknowledgements

We would like to thank Richard Ribchester for invaluable help with setting up a rig for NMJ sharp electrode recording. Nancy Bonini, Tobias Dick, Jörg Grosshans, Karen Hibbard, Fanis Missirlis,

Barret Peiffer and Alex Whitworth for generous reagent donations, and Akinao Nose and Hiroshi Kohsaka for kindly helping with *Figure 1*. We would also like to thank Jimena Berni, Alex Whitworth and members of the Landgraf lab for valuable comments on the manuscript. This work was supported by BBSRC research grants (BB/IO1179X/1, BB/M002934/1) to ML, (BB/I012273/1, BB/M002322/1) to STS and (BB/N/014561/1) to RAB.

## Additional information

### Funding

| Funder | Grant reference number | Author |
| --- | --- | --- |
| Biotechnology and Biological Sciences Research Council | BB/I01179X/1 | Matthias Landgraf |
| Biotechnology and Biological Sciences Research Council | BB/M002934/1 | Matthias Landgraf |
| Biotechnology and Biological Sciences Research Council | BB/I012273/1 | Sean T Sweeney |
| Biotechnology and Biological Sciences Research Council | BB/M002322/1 | Sean T Sweeney |
| Biotechnology and Biological Sciences Research Council | BB/N/014561/1 | Richard A Baines |

The funders had no role in study design, data collection and interpretation, or the decision to submit the work for publication.

### Author contributions

Matthew CW Oswald, Conceptualization, Resources, Data curation, Formal analysis, Supervision, Funding acquisition, Validation, Investigation, Visualization, Methodology, Writing—original draft, Writing—review and editing, Devised the project, Performed most experiments and analysis; Paul S Brooks, Investigation, Methodology, Contributed to dendrite data collection and analysis; Maarten F Zwart, Conceptualization, Writing—review and editing; Amrita Mukherjee, Investigation, Methodology, Carried out fluorescent ROS reporter experiments and analysis; Ryan JH West, Investigation, Processed specimens, Performed TEM and analysed TEM data; Carlo NG Giachello, Richard A Baines, Methodology, Writing—review and editing, Trained MO in electrophysiological methods; Khomgrit Morarach, Investigation, Contributed to dendrite data collection and analysis; Sean T Sweeney, Conceptualization, Supervision, Funding acquisition, Methodology, Writing—review and editing, Devised the project, Co-supervised the project; Matthias Landgraf, Conceptualization, Resources, Data curation, Formal analysis, Supervision, Funding acquisition, Validation, Investigation, Visualization, Methodology, Writing—original draft, Project administration, Writing—review and editing, Devised the project, Co-supervised the project

### Author ORCIDs

Matthew CW Oswald (iD) http://orcid.org/0000-0001-8586-9351
Ryan JH West (iD) https://orcid.org/0000-0001-9873-2258
Richard A Baines (iD) https://orcid.org/0000-0001-8571-4376
Matthias Landgraf (iD) http://orcid.org/0000-0001-5142-1997

### Decision letter and Author response

Decision letter https://doi.org/10.7554/eLife.39393.020
Author response https://doi.org/10.7554/eLife.39393.021

## Additional files

### Supplementary files

• Transparent reporting form

DOI: https://doi.org/10.7554/eLife.39393.018

**Data availability**

All data generated or analysed during this study are included in the manuscript and supporting files.

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
