## [Decision Letter]

Thank you for submitting your article "Reactive Oxygen Species Regulate Activity-Dependent Neuronal Plasticity in *Drosophila*" for consideration by *eLife*. Your article has been reviewed by Eve Marder as the Senior Editor, a Reviewing Editor, and two reviewers. The following individuals involved in review of your submission have agreed to reveal their identity: Konrad Zinsmaier (Reviewer #2); Stephan J Sigrist (Reviewer #3).

The reviewers have discussed the reviews with one another and the Reviewing Editor has drafted this decision to help you prepare a revised submission.

Summary:

ROS are notoriously implicated in the course of neurodegenerative disease, but much less is known how ROS intersect with "normal" mechanisms of synaptic development and plasticity. Here, the authors use genetically encoded over-activation of motoneurons to establish a paradigm for studying the role of ROS for morphological outgrowth of axonal arbors, the number of active zones, dendrite elaboration, NMJ physiology and finally behavioral output. These data are retrieved from identified motoneurons under highly standardized conditions.

The authors demonstrate that two perturbations, (1) high temperature rearing and (2) the use of temperature-activated TRPA1, both trigger NMJ outgrowth with lower active zone density. To involve ROS production, they use a mitochondrion targeted ratiometric ROS reporter *UAS-mito-roGFP2-Orp1*, which detects a small but significant increase of ROS levels upon TRPA1 activation. In order to functionally imply ROS in their activity triggered changes, they co-over-express the ROS scavenging enzymes SOD2 and catalase, which specifically "rescued" the activity induced phenotypes in the predicted direction. In a candidate approach for protein machinery to sense ROS they test for a role of DJ-1β, the fly ortholog of DJ-1 (PARK7), as a candidate ROS sensor. To evaluate a role of DJ-1β they turn into feeding paraquat (triggering the ROS NMJ response) and find that expression of a mutant form of DJ-1β that is non-oxidizable at the conserved cysteine 104 abrogates the paraquat response. They go on to involve PTEN and PI3K as downstream effectors of the DJ-1β ROS sensor. The study ends by demonstrating an interesting adaptation and rebound effect regarding larval locomotion, which they observe by combining chronic high temperature rearing and acute temperature shifts of the larvae. They show that the mutated DJ-1β ROS sensor can to a degree abrogate these adaptive/plastic effects.

Reviewers generally agree that the findings in the paper are both interesting and important. A highlight of the work is the extension of work on ROS to an in vivo system where neurodevelopment of synaptic connections and neuronal morphology can be easily and clearly quantified. A number of issues were raised in the review process that, if addressed, would solidify the major conclusions of the study.

Essential revisions:

1) The effect of the DJ-1β Δ93 null mutation on the ROS-induced (DEM or paraquat) increase in bouton number (Figure 3A) is key for the main conclusion of the present study. To ensure that this effect is specific to the loss of DJ-1β, it should be genetically rescued by expression of normal DJ-1β in Δ93 null mutants.

2) What is the effect of expressing only DJ-1βC104A on the ROS-induced increase in the number of boutons (Figure 3A)?

3) Concerns were raised regarding the genetic controls for anatomical experiments. In some instances, control genotypes appear to be either Oregon R or unspecified. In instances where comparisons are made between genotypes expressing transgenes, then the appropriate controls should include the heterozygous driver and/or UAS-transgene in the absence of a driver. Please include these controls where appropriate and ensure that statistically significant changes remain robust.

4) Given the variance in the electrophysiological recordings, sample sizes of 4 or 5 synapses seem quite small. The effects seem robust, but it would be important to extend these sample sizes to ensure that the effects are not simply due to sampling errors from overlapping data distributions with large variance.

5) More explanation is necessary regarding some of the major conclusions. How can we explain that catalase is needed for "homeostatic maintenance" but not DJ-1β? Does this mean that though catalase and DJ-1β manipulate parameters such as bouton number and active zone number similarly, their impact on release is uncoupled from these parameters?

6) There was discussion about the assertion that crawling behavior is homeostatically stabilized through some type of network process. Here, the reviewers recommend more careful discussion. Is normal crawling speed an emergent property of homeostatic mechanisms acting at a cellular level? It is hard to imagine that the animal has a fundamental crawling speed that is sensed and used as feedback information to instruct neural function or development. To support such a conclusion, it might be necessary to confront the animals with more challenging behavioral paradigms that specifically stress the ability of the animals to crawl, rather than acting primarily at the cellular level. Without such an experiment, a more measured discussion of homeostasis and potential mechanisms that might underlie the observed stabilization of crawling speed seems appropriate.

---

## [Author Response]

Essential revisions:1) The effect of the DJ-1ß Δ93 null mutation on the ROS-induced (DEM or paraquat) increase in bouton number (Figure 3A) is key for the main conclusion of the present study. To ensure that this effect is specific to the loss of DJ-1ß, it should be genetically rescued by expression of normal DJ-1ß in Δ93 null mutants.

We carried out the suggested experiment and find that neuronal *UAS-DJ-1ß* expression, in a DJ-1ß null mutant background, leads to increased bouton number as expected due to inhibition of PTEN. Critically, this phenotype is exacerbated in the presence of DEM, suggesting that oxidised DJ-1ß more potently inhibits PTEN to drive increased bouton number (see Figure 3—figure supplement 3).

This observation is consistent with a neuronal requirement for DJ-1ß, and that the inhibition of PTEN by DJ-1ß is modified by DJ-1ß oxidation state, consistent with data presented by others and within this manuscript.

2) What is the effect of expressing only DJ-1ßC104A on the ROS-induced increase in the number of boutons (Figure 3A)?

This is a good suggestion. Expression of mutant DJ-1ß-C104A (non-oxidisable on the conserved cysteine C104) has a dominant effect since DJ-1ß has been shown to act as a dimer. We find that this manipulation leads to a similar phenotype as a DJ-1ß null mutant, namely a lack of NMJ growth adjustment normally seen in response to DEM exposure (see Figure 3A).

3) Concerns were raised regarding the genetic controls for anatomical experiments. In some instances, control genotypes appear to be either Oregon R or unspecified. In instances where comparisons are made between genotypes expressing transgenes, then the appropriate controls should include the heterozygous driver and/or UAS-transgene in the absence of a driver. Please include these controls where appropriate and ensure that statistically significant changes remain robust.

We have adjusted figure legends to enhance clarity on control genotypes. As requested, we have carried out a battery of control experiments, testing for possible effects of the UAS-responders, e.g., as might be expected from ‘leaky’ transgene expression. None of the 11 UAS-responder controls (no GAL4) in this study showed any phenotypes, underlining that the effects seen are due to GAL4 mediated expression of the UAS-responder transgenes, rather than non-specific leakiness. However, please note that we were unable to carry out one control, due to the relevant stock having died and us having been unable to source a new copy until now; this is UAS-DN-PI3Kinase / + (without GAL4). We did not want to delay resubmission much longer (flies need re-balancing, then crossing), so decided to submit the paper at this stage; however, if deemed necessary we can carry out that control as well.

4) Given the variance in the electrophysiological recordings, sample sizes of 4 or 5 synapses seem quite small. The effects seem robust, but it would be important to extend these sample sizes to ensure that the effects are not simply due to sampling errors from overlapping data distributions with large variance.

As suggested, we have carried out additional experiments to bolster sample sizes. This has led to one minor change, namely revealing statistically significant effects on quantal content under conditions of over-activation and ROS quenching by mis-expression of UAS-Catalase, as one might have expected (see Figure 5C).

5) More explanation is necessary regarding some of the major conclusions. How can we explain that catalase is needed for "homeostatic maintenance" but not DJ-1ß? Does this mean that though catalase and DJ-1ß manipulate parameters such as bouton number and active zone number similarly, their impact on release is uncoupled from these parameters?

Yes, that is precisely our interpretation, that the structural changes to synaptic terminal size and synapse number appear to be at least in part uncoupled from changes to synaptic transmission; and that we see two separable aspects of synaptic transmission: the amplitude of mEJPs being sensitive to DJ-1ß oxidation, while the amplitude of eEJPs is impacted by ROS sensitive to Catalase expression.

Structurally, the regulation of bouton number/NMJ growth and active zone number is sensitive to both, Catalase levels and DJ-1ß oxidation at Cystein104. At this point in time we simply cannot determine to what extent either of these structural changes is regulated separately from one or other transmission aspect; we have not yet seen ROS-mediated changes in synaptic transmission without a parallel impact on NMJ structure, or vice versa. One way of thinking about this issue is to consider ROS as spatio-temporally well controlled signals, much like phosphorylation events. In such a scenario the location and nature of ROS generated, be that in mitochondria or at the cell membrane via NAPDH oxidases, would be important. That could allow for finely controlled redox regulation of plasma membrane associated signalling pathways (e.g., PTEN-PI3Kinase mediating growth and potentially non-autonomous signals to target muscles) versus cytoplasmic cytoskeletal/active zone cytomatrix modifications (e.g., mediating changes in active zone number and/or synaptic transmission).

We have attempted to clarify this point in the manuscript (Materials and methods section).

6) There was discussion about the assertion that crawling behavior is homeostatically stabilized through some type of network process. Here, the reviewers recommend more careful discussion.

We have edited the relevant section and hope to have struck a clearer, more considered tone.

Is normal crawling speed an emergent property of homeostatic mechanisms acting at a cellular level? It is hard to imagine that the animal has a fundamental crawling speed that is sensed and used as feedback information to instruct neural function or development. To support such a conclusion, it might be necessary to confront the animals with more challenging behavioral paradigms that specifically stress the ability of the animals to crawl, rather than acting primarily at the cellular level. Without such an experiment, a more measured discussion of homeostasis and potential mechanisms that might underlie the observed stabilization of crawling speed seems appropriate.

We agree. A homeostatic system requires some form of feedback – for the locomotor network such feedback will most likely come, at least in part, through proprioceptive sensory neuron feedback.